# Understanding Atherosclerosis Pathophysiology: Can Additive Manufacturing Be Helpful?

**DOI:** 10.3390/polym15030480

**Published:** 2023-01-17

**Authors:** Joana Henriques, Ana M. Amaro, Ana P. Piedade

**Affiliations:** CEMMPRE-Department of Mechanical Engineering, University of Coimbra, 3030-788 Coimbra, Portugal

**Keywords:** atherosclerotic plaques, mechanical properties, standardization, additive manufacturing

## Abstract

Atherosclerosis is one of the leading causes of death worldwide. Although this subject arouses much interest, there are limitations associated with the biomechanical investigation done in atherosclerotic tissues, namely the unstandardized tests for the mechanical characterization of these tissues and the inherent non-consensual results obtained. The variability of tests and typologies of samples hampers direct comparisons between results and hinders the complete understanding of the pathologic process involved in atherosclerosis development and progression. Therefore, a consensual and definitive evaluation of the mechanical properties of healthy and atherosclerotic blood vessels would allow the production of physical biomodels that could be used for surgeons’ training and personalized surgical planning. Additive manufacturing (AM), commonly known as 3D printing, has attracted significant attention due to the potential to fabricate biomodels rapidly. However, the existing literature regarding 3D-printed atherosclerotic vascular models is still very limited. Consequently, this review intends to present the atherosclerosis disease and the consequences of this pathology, discuss the mechanical characterization of atherosclerotic vessels/plaques, and introduce AM as a potential strategy to increase the understanding of atherosclerosis treatment and pathophysiology.

## 1. Introduction

Cardiovascular diseases are one of the most common causes of death worldwide, accounting for over 17 million deaths per year [1]. With static projections pointing to >23.6 million deaths in 2030, it is essential to thwart these numbers and solve this epidemiologic problem [1]. Atherosclerosis and aneurysms are the most frequently occurring vascular diseases and both are characterized by significant remodeling of the vessel wall structure and function [2]. The architectural arrangement of the vessel walls, as well as changes in quantitative ratios of components and unbalanced biochemical reactions, are some of the severe consequences that such diseases may be associated with. Ultimately, these conditions may interfere with the biomechanical properties of blood vessels and, consequently, affect the physiological and normal performance of these vessels. Therefore, it is crucial to increase the knowledge of vascular disease pathophysiology and, thus, develop new therapeutic strategies [3].

While aneurysms are related to a permanent and irreversible localized dilation of a blood vessel [4], atherosclerosis results from the accumulation of lipids in the vessel wall, which leads to the invasion of exogenous cells and compounds into the tunica intima (Figure 1), causing chronic inflammations [5]. Recruitment of smooth muscle cells from the tunica media (Figure 1) to the innermost layer (intima) leads to a thickening of the intima layer and an increased synthesis of fibrotic components and calcium salts deposition [2].

This stage of atherosclerosis lesion, also known as atherosclerotic plaque (Figure 1), can vary in density and cellularity [5], including calcified areas, fibrotic regions, lipid-rich nature, intraplaque hemorrhage, and a thin or thick fibrous cap [2]. Advanced stages of atherosclerosis, characterized by plaque rupture and subsequent release of highly thrombogenic material and lipids into the blood, may reduce or block the flow of oxygen-containing blood, leading to oxygen deficiencies in tissues, which eventually could result in significant clinical emergencies such as myocardial infarction and stroke [5].

Consequently, vascular modeling is essential to progress the understanding of atherosclerosis development and find adequate therapeutic interventions. Detailed mechanical characterization is vital in designing and developing successful vascular models so they can accurately mirror the mechanical attributes of blood vessels [7]. Predictable and translatable models include “the crosstalk between essential cellular and tissue components, specifically endothelial cells (ECs), smooth muscle cells (SMCs), extracellular matrix (ECM), and blood constituents’ underflow” [3]. Such models should be capable of recapitulating the pathophysiologic functions of atherosclerotic blood vessels, thus, helping to understand how the vascular disease progresses, the consequences of its development, and how it can be solved.

After knowing the mechanical properties of normal and diseased blood vessels, creating the vascular model is the next step. To do so, AM has emerged as a popular tool for understanding the pathophysiology and function of patient-specific vascular diseases. Using 3D printing can accurately reproduce vascular anatomy, including bifurcations and curvatures of vascular networks [3]. The speed, design freedom, and variety of materials in AM allow realistic simulation of vascular pathologies, helping to reduce the number of animal experiments in preclinical research and development [8].

This review aims to give a perspective on modeling atherosclerosis pathophysiology via AM. First, the vascular system will be presented, namely the different types of blood vessels in the human body and the differences and similarities between them. Then, an overview of the mechanical properties of normal blood vessels will be given. Secondly, atherosclerosis disease will be contextualized, including plaque classification and occlusive complications associated with this disease. Mechanical properties of atherosclerotic components and atherosclerotic blood vessels are also presented, including the most commonly used techniques to mechanically characterize those tissues. Finally, the 3D printing strategy used to produce diseased vascular models will be described, and the available models in the literature will be discussed.

## 2. Mechanical Properties of Normal Blood Vessels

As aforementioned, blood vessels are composed of a combination of different cell layers that should be able to withstand and propagate the physiological forces that are applied to them, namely blood flow (tangential force), blood pressure (radial and longitudinal forces), and surrounding tissue forces (or tethering, which leads to longitudinal forces) [7,9].

Nevertheless, the mechanical properties of a material depend not only on its composition but also on its structure and ultrastructure. For instance, the properties of an artery depend not only on how much collagen it has, but also on how the collagen fibers are arranged in the tissue [10]. Moreover, since the composition and structure of blood vessels vary along the arterial tree, it is challenging to address the mechanical properties of all blood vessels of the vascular system. Moreover, differences in sample sizes, test methods, tissue sources, and sample processing methods can originate a wide variation in mechanical properties [11].

Due to the multi-layered structure of vessel walls, blood vessels have composite material characteristics that confer unique mechanical properties. Such unique features include anisotropy, non-linearity, viscoelasticity, and compliance. Anisotropy is due to the geometrical arrangement of fibers in the circumferential direction, making blood vessels stronger in the fiber direction than in the perpendicular direction [7].

Non-linearity is associated with elastin and collagen fibers within the vessel wall, and it is a critical property protecting blood vessels from rupturing when there is a strong and instant increased blood demand (i.e., sportive activity or during a burst of laughter or crying) [11]. At lower pressures (<80 mmHg), the mechanical behavior is dominated by elastic components of the vessel wall that are less stiff and more elastic [7]. In this early stage of radius expansion, blood vessels show a low circumferential Young’s modulus (usually 20–50 kPa), indicating good flexibility [11]. As the pressure increases, the fibers start to straighten gradually, and at the lower value of physiologic pressure (80 mmHg), elastin fibers are nearly straight. The load transits between the elastin and collagen fibers in the physiologic pressure range (80–120 mmHg). In this range, elastin fibers are stretched, and the collagen fibers are continuously straightened [7]. For 30–60% strain expansions, Young’s modulus (E) increases sharply (100–200 kPa), constraining further lumen expansion [12]. For blood pressures higher than 120 mmHg, elastin and collagen fibers are fully stretched. In this range of higher pressures, the mechanical behavior of the vessel wall is dominated by rigid collagen fibers that are stiffer and capable of protecting the blood vessels against damage or rupture when blood pressure is very high [7].

Another important mechanical feature of the blood vessel is the viscoelastic property. Due to this characteristic, when the tissue is stretched, and strain is kept constant, the induced stress decreases with time; contrarily, when the applied stress is maintained constant, the tissue continues to deform. These combined processes lead to hysteresis, which is related to a delayed response when the cyclic inflation–deflation stresses are applied. The difference between the stress–strain curve obtained in the loading process and the curved resulting from the unloading process originates a loop profile. The area of the loop corresponds to the energy lost during the distention of the wall, which helps attenuate pressure pulses that propagate along the arteries [7].

Finally, compliance is the mechanical property involved in the propagation of pulsatile blood flow [7]. It is an indirect measure of the capacity of the arteries to store blood, and it represents their ability to convert the pulsatile flow that comes from the aorta to a continuous flow that will reach the capillaries [7]. Defined as the elastic deformation of a material in response to pressure, compliance associated with a vascular structure corresponds to the changes in the dimensions of the tube that occur in response to pressure variations within the lumen [11].

Although less referred than the other mechanical properties, burst pressure is also a critical biomechanical feature when it comes to evaluation of the performance of blood vessels. It stands for the maximum pressure that a vascular vessel can withstand before an acute leak occurs and it fails [11], and its value is given by the expression: f = Pd/2t, where f, P, d and t are the maximum force, burst pressure, diameter, and wall thickness, respectively [11]. Observing the expression, as the wall thickness decreases and the diameter increases, the burst pressure decreases.

Some studies have already been done regarding the mechanical properties of blood vessels and their main wall components (collagen, elastin, and SMCs). In Table 1, some examples are listed.

## 3. Atherosclerotic Disease

Vascular diseases result from structural and functional changes in blood vessels that cause blood flow disturbances and affect the blood supply to the organs and tissues [3,20]. Arterial stenosis is a vascular disease characterized by the narrowing of the blood vessel lumen [21]. By disturbing blood flow, narrowed lumen precludes the adequate irrigation of perfused organs, which can be associated with myocardial infarctions, ischemic attacks, and stroke [22]. Stenosis can either be caused by extrinsic factors, such as external compression triggered by aneurysms and tumors or associated with intrinsic aspects currently related to atherosclerosis. Indeed, advanced atherosclerosis is the leading cause of vascular stenosis [21].

Atherosclerosis is a chronic systemic inflammatory disease [23] associated with plaque accumulation within the vessel wall and characterized by intramural lipid retention coupled with inflammation and dyslipidemia [20]. Inflammation is, indeed, a critical event involved in the pathological process of atherosclerosis [24]. Generally, atherosclerosis spreads out in large and medium thick-walled systemic arteries, specifically near or into branching regions, or in vascular regions characterized by a low wall shear stress, where the circulation time of lipoproteins and cells is long enough so they can easily sediment and cross the arterial wall. This implies that atherosclerosis preferentially occurs in stenotic, curved, and branched arterial regions, where disturbed flows are generated [25].

Just as with every vascular disease, the initial stage of atherosclerosis is triggered by a disturbance in blood flow, caused mainly by the exposition to different cardiovascular risk factors, external stimuli, or damage, that leads to a drastic change in the expression of regulatory molecules and a subsequent unbalanced release of pro- and anti-inflammatory factors, which in turn activates the endothelium [23,24].

Endothelial activation is nothing more than a loss of normal homeostatic properties of the endothelium, and it is a phenomenon associated with several activities: (1) the recruitment of immune cells, like monocytes and leukocytes; (2) the secretion of inflammatory cytokines and growth factors; and (3) the release of prothrombotic mediators and platelet activation that confers a procoagulant status instead of a normal anticoagulant response [3]. Parallelly, the permeability of the endothelial monolayer is altered, favoring the transmigration of leucocytes within the subendothelial space [23]. The persistence of the inflammatory response within the subendothelial space instigates the migration and proliferation of SMCs that become intermixed with the inflammation area, progressively thickening the vessel wall [3,23]. This pro-inflammatory environment also promotes the oxidation of low-density lipoproteins (LDLs) previously accumulated in the vessel wall (specifically in the tunica intima) due to endothelial activation. Oxidized LDL (oxLDL) are engulfed by monocyte-derived macrophages through phagocytosis, and the retention of oxLDL in the cytoplasm transforms macrophages into lipid-filled foam cells [26], constituting the core of atheroma [23]. The necrotic lipid core comprises foam cells, cellular debris, and extracellular lipids and is surrounded by a fibrous cap. The fibrous cap is, in turn, composed of SMCs and matrix proteins like collagen, elastin, and proteoglycans [23].

Calcified plaques are related to the advanced stages of atherosclerosis and are traditionally regarded as stable atheromas that cause stenosis. This calcification process occurs in the vascular intima and is analogous to the multistep process of bone formation [12]. Calcified deposits result from a mineralized matrix generated by vascular SMCs and other cells (adventitial myofibroblasts) after they undergo osteoblastic differentiation [12]. After the increased synthesis of fibrotic components, deposition of calcium salts may occur.

Therefore, besides causing structural modifications in the blood vessel through decreasing lumen diameter, atherosclerosis also triggers the hardening of the vessel wall, as happens in the calcified deposits and fibrotic plaques [3,26]. Atherosclerotic lesions have, thus, two significant consequences: (1) blood vessel lumen narrowing, thus compromising the blood supply to vessels downstream; and (2) the alteration of flow in the affected vessel, which changes the loading, shear stress, and pressure of the fluid on the vessel wall [27]. These factors can affect both the further progression of the lesion and its tendency to fracture and rupture, and the rupture of the cap of vulnerable plaque is an undesired consequence of atherosclerosis, being the cause of the acute manifestations of this disease, like stroke, ischemic attacks, and myocardial infarctions [22]. In fact, contrary to expectation, the mortality associated with atherosclerotic disease is not simply correlated with stenosis degree. However, it is strongly affected by plaque vulnerability, herein defined as the tendency of a plaque part to break off the wall and block smaller vessels downstream [27]. Identifying the rupture-prone plaques is essential to know how to treat the damaged vessel and prevent the disease’s progression.

Usually, vulnerable plaques appear in advanced stages of atherosclerosis and are strongly dependent on plaque composition. For instance, lesions with large lipid pools and thin fibrous caps are more prone to rupture, resulting in clot-promoting materials exposed to the lumen [20]. Moreover, how different plaque components are combined influences the macroscopic material plaque properties. This way, knowing the different stages of atherosclerosis and the classification of atherosclerotic plaques, it is possible to identify vulnerable plaques, recognize the components affecting plaque vulnerability, and understand how the mechanical behavior of the blood vessels is affected by the different types of plaques [22].

The American Heart Association has adapted and adopted a plaque classification scheme composed of eight atherosclerotic lesion types representing the successive stages of atherosclerosis [22,28].

The initial stages of atherosclerosis consist of isolated macrophage foam cells containing lipid droplets [20]. These early lesions and fatty streaks (type I and type II lesions, respectively) evolve to intermediate lesions defined as pre-atheroma (type III lesions), characterized by the deposition of extracellular lipids between the SMCs, forming lipid pools within the intima. These lipid pools are responsible for several processes: (1) the disruption of the cellular structure of the vessel, (2) the break of elastic fibers; and (3) the thickening of the intima, keeping the tunica media and tunica adventitia unaffected [20,22].

From pre-atheroma, atherosclerotic plaque develops into atheroma and fibro-atheroma (type IV and type V lesions), comprising a lipid core separated from the lumen by a proteoglycan-rich cap, as a response to the intimal disorganization. While large amounts of collagen and SMCs do not characterize atheroma lesions, in fibro-atheroma lesions, there is an increased synthesis of fibrous tissue composed of SMCs and collagen. Generally, in both situations, media and adventitia are disarranged.

After the formation of (fibro-) atheroma, a rupture lesion (type VI lesion) could appear. This lesion is associated with a lesion surface defect or intra-plaque bleeding. The progression of atherosclerosis leads to calcified lesions (type VII) or fibrotic lesions (type VIII). In calcified lesions, intima is fibrous with calcifications and some lipid depositions can be present, whereas in fibrotic lesions, tissue is fibrous and the lipid core is not present [22].

### 3.1. Occlusive Complications of Atherosclerosis

Atherosclerosis is an intimal disease that can be seen as an occlusive or stenotic condition due to the narrowing of the vessel lumen. This type of event affects the mechanical properties of the vessel that has been occluded [29].

To study the effect of an occlusive event on the biomechanical behavior of a blood vessel and the disturbances of blood flow across the stenosis, Tsatsaris et al. (2004) [30] have induced stenosis in thoracic aorta of pigs by circumferential symmetrical constriction of those vessels. Results have shown that the aortic wall of stenotic vessels became stiffer, particularly at high strains. In addition, the authors have histologically analyzed aortic vessels and compared stenotic groups with non-stenotic groups. They observed that while the percentage of elastin fibers remained the same, the percentage of collagen fibers increased considerably. These findings were explained by the reverse flow in the post-stenotic region, which increases the production of collagen fibers and makes the aortic wall stiffer. Tsatsaris and co-workers have concluded that a stenotic event accelerates blood flow when it passes the strangulated area, causing a high and fast reverse flow. Consequently, reversed and accelerated blood flow enhances the production of collagen fibers by fibroblasts to help the aortic wall withstand the increased shear load, so the aorta becomes stiffer [30].

Besides affecting the mechanical behavior of blood vessels, occlusive consequences associated with atherosclerosis are also related to other pathologic complications. As the atheroma enlarges, it can suffer erosion and rupture, causing a sudden thrombotic occlusion. If this thrombogenic atheroma, also known as “unstable plaque”, appears in cerebral arteries, coronary arteries, or in arteries of the leg, cerebral or myocardial infarction may occur [29].

### 3.2. Mechanical Properties of Atherosclerotic Vessels

#### 3.2.1. Mechanical Properties of Individual Components of Atherosclerotic Plaque

As the atherosclerotic plaque develops, its composition varies progressively. The main components found in most plaques are lipid pools, SMCs, collagen, elastin, and calcified deposits [22]. Depending on the plaque classification, the amount of each element in tunica intima is different, implying different mechanical properties. In order to be capable of reproducing an atherosclerotic blood vessel with different degrees of plaques, it is essential to know the mechanical properties of the materials that compose those plaques. Mechanical properties are critical in the choice of material that will constitute the phantom. However, they will also affect the response of the mimicking-vessel model to the forces and strains applied to simulate a physiological condition—having that in mind, it will be possible to obtain a pathological model that accurately mimics the diseased blood vessel.

Atherosclerotic lipid pools mainly comprise phospholipids, cholesterol esters, cholesterol crystals, and other lipids. Although there are no available experimental data regarding the mechanical behavior of lipid pools [22], Loree et al. (1994) [31] have developed synthetic models of lipid pools with various lipid compositions subjected to shear experiments, and they have found that cholesterol crystals lead to stiffening of the lipid pool. Depending on the number of cholesterol crystals, shear experiments have yielded stiffness moduli ranging from 50 to 300 Pa [31].

SMCs, specifically vascular smooth muscle cells (VSMCs), play an important role in the formation and progression of atherosclerosis [24], thus, strongly affecting vessel and plaque biomechanics. The early stages of atherosclerotic plaque formation are characterized by a change of VSMCs’ phenotype from a contractile to a synthetic phenotype and, consequently, by a decrease in the expression of contractile proteins. This phenotype change enhances the expression of growth factors that allows the migration and proliferation of VSMCs from tunica media to intima, which results in plaque formation. Additionally, as the synthetic VSMCs are less stiff than contractile VSMCs, the diseased blood vessel will be softer [32], with reported stiffness values as low as 2.6–2.9 kPa, obtained for rat aortas [33]. In advanced stages of atherosclerosis, VSMCs continuously suffer apoptosis and release large amounts of matrix metalloproteinase that degrade the extracellular matrix and thin the fibrous cap of plaques [24].

Collagen is another component largely influencing the overall mechanical behavior of plaque. The great dominance of collagen fibers makes the vessel wall harder and stiffer, with lower elastic properties. Due to their high strength, collagen fibers make vessel walls resistant to rupture [2]. Although collagen enhances the stiffness value of vessels by itself, this protein’s contribution to the intima’s load-bearing capacity depends on fibers distribution and their crosslinking [29]. Bruel et al. (1998) [34] have used rats to study the influence of crosslinking on the mechanical stability of thoracic aorta, and they have found that by inhibiting the formation of crosslinks in collagen fibers, the maximum stiffness of aortic samples is significantly reduced when compared to the controls. 

Since there are only very low amounts of elastin in atherosclerotic intima, elastin is considered a minor contributor to the mechanical behavior of atherosclerotic tissue [22]. Nevertheless, arterial elastin has an estimated elastic modulus in the order of 0.88–1.14 MPa [14], mostly contributing to the elastic behavior of the blood vessel.

Calcified inclusions in atherosclerotic intima are responsible for strongly affecting the mechanical properties of a plaque, as they can be stated as rigid inclusions. As referred by Akyildiz et al. (2014) [22], calcified samples showed stiffness values between 100 MPa and 21 GPa, undoubtedly stiffening materials but contributing heterogeneously to the mechanical properties of the lesion. Indeed, morphological aspects of atherosclerotic tissue, such as collagen fiber orientation and calcified tissue geometry, also considerably influence the mechanical response of vascular tissues to tensions applied to them [35].

Holzapfel et al. (2004) [5] have studied the mechanical properties of tissue components in human atherosclerotic plaques. They used 107 samples from nine human high-grade stenotic iliac arteries, and the different tissue types were characterized through histological analysis. They obtained seven different types of tissues that underwent cyclic quasistatic uniaxial tension tests in axial and circumferential directions. Most samples showed an anisotropic and highly nonlinear tissue; however, calcified samples presented a linear property with about the same stiffness as observed for the adventitia in high-stress regions. The stress and stretch values at calcification fracture were smaller (179 ± 56 kPa and 1.02 ± 0.005) than for the other tissue components. Of all intimal tissues investigated, the lowest fracture stress occurred in the circumferential direction of the fibrous cap (254.8 ± 79.8 kPa at a stretch 1.2 ± 0.1). While the adventitia demonstrated the highest mechanical strength on average in both tensile directions, the non-diseased media showed the lowest values [5].

On the other hand, Carrera et al. (2014) [36] presented the Young’s modulus of the tissues that compose atherosclerotic plaques, and in their report, they present that calcified deposits are the stiffest tissues (E = 12 MPa) and lipid pools are the softest (E = 0.1 MPa). Fibrotic media has a stiffness value that is five times higher than the un-diseased media (E = 1 MPa), and the adventitia and fibrous cap have a similar stiffness value (E = 2.5 MPa). These values were obtained via modulation, making the consideration that tissue materials have linear isotropic properties.

In contrast, particularly the effect of lipid presence, in the viscoelastic properties of blood vessels, Pynadath et al. (1977) [37] have studied the changes in the dynamic Young’s modulus of rabbit aorta concerning the cholesterol and ester cholesterol content of these tissues. The investigation was done in male rabbits divided into two groups, one of which received a 1.5% cholesterol diet for six weeks. Animals were sacrificed every two weeks to obtain different degrees of plaque development. After extraction, one-half of each aorta was used to determine cholesterols content, and the other half was used to assess the viscoelastic properties. The dynamic Young’s modulus was determined using a dynamic viscoelastometer by applying a minor sinusoidal strain on the tissue segment and the subsequent follow-up of the corresponding response in the load level (measurements were done at a frequency level of 110 Hz). Results have shown that the cholesterol diet was effective in inducing aortic lesions, with mild lesions appearing at the end of 2 weeks and severe lesions at the end of the 6 weeks. Although there were no changes in the longitudinal dynamic Young’s modulus (E_long_) of the aortic tissue, the tangential dynamic Young’s modulus (E_tang_) of the aortas was found to be very much influenced by the cholesterol diet, and thus, significantly affected by atherosclerosis. While no aortic lesions presented an E_tang_ value of 290 kPa, mild and severe aortic lesions showed values of 330 kPa and 400 kPa, respectively. This tendency was explained by the fact that the filtration of lipids into the medial interlamellar elastic tissue-collagen web may cause the loss of elasticity of the aortic wall [37].

Finally, Rezvani-Sharif et al. (2019) [15] have determined the local Young’s modulus of the wall and plaque components of healthy, mildly diseased, and advanced atherosclerotic human abdominal aortas. Atherosclerotic plaque components were histologically stained and classified as calcification zone, fibrous cap, or lipid pool. The Young’s modulus of aortic wall lamellae and plaque components were derived from the force spectroscopy mode of atomic force microscopy. Authors have found that elastin lamellae become less stiff as atherosclerosis develops (reduction of 18.6%), and the stiffness of interlamellar zones increases significantly with the formation and development of atherosclerotic plaques (50%). This stiffening in the interlamellar zone can be related to the superior synthesis of collagen fibers by SMCs in response to non-physiological strain and stress ranges in atherosclerotic vessels. Young’s moduli were also different between plaque components. While the calcification zone was the stiffest (103.7 ± 19.5 kPa), the lipid pool was the softest (3.5 ± 1.2 kPa); the fibrous cap presented E values of 15.5 ± 2.6 kPa [15].

#### 3.2.2. Strategies to Measure Mechanical Properties of Atherosclerotic Blood Vessels

The mechanical properties of blood vessels are affected by a series of combined factors that are intimately related to the cells that compose vessels’ layers, namely cell layer thickness, the volume of cells, their molecular orientation, surrounding tissues, as well as genetic characteristics, lifestyle, and age [9,38,39]. Due to the diversity of cell types, topologies, and unique structures, in addition to the small dimensions of blood vessels and their different natures and sources, determining the mechanical properties of all types of blood vessels has been a tricky challenge.

Nevertheless, and as already mentioned, the mechanical properties of atherosclerotic blood vessels are crucial to generating a reliable vascular model that mimics a diseased vessel. In this sense, many efforts have been made to assess the mechanical properties of atherosclerotic plaques and/or atherosclerotic blood vessels, and there are numerous methods to do that characterization [40]. However, the wide range of techniques to characterize such biological tissues mechanically thwarts a consensus about the mechanical properties of blood vessels, either non-diseased or atherosclerotic vessels.

Studying the dynamic vascular changes of the vessel wall is one of the possibilities to evaluate the elastic properties of blood vessels, specifically arteries (e.g., carotid arteries). Indices like distensibility, compliance, Young’s modulus, pressure-strain elastic modulus, and stiffness index are commonly used to estimate vessels’ mechanical properties and evaluate functional changes in vessel walls [41]. This strategy has been appointed as a possible way to detect atherosclerotic damage early and, therefore, to treat pre-symptomatic patients with atherosclerosis disease [42] effectively. Functional changes assessment is achievable by measuring the vascular diameter, wall thickness, cross-section changes, and blood pressure variability [41].

Table 2 lists the indices that can be used to express the local elasticity or stiffness of blood vessels as a function of dynamic changes in these tissues during a cardiac cycle. The most basic measure is absolute distention, which can be expressed in terms of diameter (ΔD) (Equation (1)) or cross-sectional area (ΔA) (Equation (2)) variations. Normalizing this value to the diastolic diameter produces the oft-reported strain (Equation (3)) (typically reported as a percentage of diastolic diameter) [20]. However, these indices do not account for the blood pressure exerted on the vessel, which is a factor affecting vessel compliance. Instead, indices like Peterson’s pressure-strain elastic modulus ($E_{p}$) (Equation (4)), *E* (Equation (7)), distensibility (D) (Equation (5)), and distension coefficient (DC) (Equation (6)) normalize the changes in cross-section area (or diameter) by pulse pressure ($\Delta P=P_{s}-P_{d}$, where $P_{s}$ and $P_{d}$ are systolic and diastolic blood pressure respectively). The β-stiffness index (Equation (8)), in turn, accounts for the effect of blood pressure by taking the natural logarithm of the systolic to diastolic blood pressure ratio and dividing by the strain [20]. 

Arterial stiffness can also be evaluated through pulse wave velocity (PWV) that measures the propagation of the pressure wave along the arterial tree. PWV is calculated (Equation (9)) by measuring the time taken for a pressure pulse to travel between two set points (commonly between the carotid and femoral artery, because they are superficial and easy to access). When atherosclerosis is present, the stiffness of large arteries is increased [43].

To understand the effect of atherosclerosis on the mechanical behavior of blood vessels, Mokhtari-Dizaji et al. (2005) [42] have examined the differences in elasticity of the common carotid artery under normal and atherosclerotic conditions through direct estimating of static pressure using the energy conservation law. According to the authors, while Young’s modulus essentially measures the intrinsic elastic properties of a given material, independent of the thickness of the material, $E_{p}$ considers the amount and the configuration of material, representing the overall arterial stiffness, limiting the pulsatile expansion of the artery [42].

In this study, all subjects (128 men with an average age of 66 ± 11, of which 55 are healthy with no history of cardiovascular disease, cerebrovascular disease, hypertension and/or diabetes, and 73 patients with angiographic documented coronary artery disease) underwent color Doppler ultrasonography to determine diameter narrowing and to divide the subject into three different groups: normal (no diameter narrowing), mild stenosis (narrowing of less than 40%), and severe stenosis (diameter narrowing of more than 40%). The authors have concluded that as the stenosis degree becomes more severe, the diameter of common coronary decreases, which is under the expectation and associated with the increase in wall thickness. Regarding arterial strain, authors have found that the normal group (9.4 ± 3.5%) had the highest value, and the patients with severe stenosis (5.8 ± 2.8%) had the lowest value (mild stenosis group presented an arterial strain of 7.7 ± 3.2%). The decrease in strain is due to the change in the collagen/elastin ratio associated with atherosclerosis, so increasing this ratio decreases the arterial strain. 

In turn, the static pressure–strain elastic modulus ($E_{ps}=\frac{\Delta P_{s}D_{d}}{\Delta D}=\frac{\frac{1}{2}\rho \left({V_{1}}^{2}-{V_{2}}^{2}\right)D_{d}}{\Delta D}$) was greater in patients with severe stenosis (6.5–7.0 kPa) than those with mild stenosis (~3.9 kPa), and no significant difference between patients with mild stenosis and the normal group (~1.9 kPa) was observed. This could be explained by the fact that when a vessel stiffens by a pathological process such as atherosclerosis, the static pressure ($\Delta P_{s}$) throughout the cardiac cycle rises and falls extraordinarily, leading to higher $E_{ps}$ values when severe stenosis (advanced atherosclerosis) is present [42]. 

In another study, Mokhtari-Dizaj et al. (2006) [41] used a similar method to calculate arterial $E_{p}$, stiffness, *E*, *D*, and compliance (defined as $\frac{A_{s}-A_{d}}{P_{s}-P_{d}}$). The investigation was done in 60 healthy human patients, 41 patients with mild stenosis (lumen narrowing of less than 40%), and 44 patients with severe stenosis (more than 40% of lumen narrowing). Changes in arterial cross-section of the common carotid artery were grabbed via ultrasound images (30 frames/s), and the diameter and intima-media wall thickness were estimated by color Doppler imaging throughout two cardiac cycles. The motion estimation from the sequences of color Doppler images was performed by optical flow tracking technique. The authors have concluded that with the progression of stenosis, there was an increase in the arterial $E_{p}$ (from 79.85 kPa to 160.34 kPa), stiffness index (from 6.12 to 8.24), and *E* (from 384.26 kPa to 577.23 kPa), and a decreasing in values of the arterial strain (from 9.9% to 5.2%), *D* (from 29.4 × 10^−3^ kPa^−1^ to 16.0 × 10^−3^ kPa^−1^), and compliance (from 14.10 mm^2^/Pa to 6.55 mm^2^/Pa). They also found no significant difference between groups in measured blood pressure because severe stenosis patients usually use hypertension drugs, which normalizes blood pressure in these pathological conditions. Pathologic groups presented an arterial strain value smaller than that obtained in the healthy group due to the increase in arterial wall thickness and the collagen/elastin ratio change associated with atherosclerosis. The stiffness values were associated with the chosen method to obtain the systolic blood pressure and ΔP that had led to overestimated values [41].

Van Popele et al. (2001) [44] have investigated the effect of atherosclerosis on arterial stiffness of the common carotid artery. The study was performed on 3000 subjects aged between 60 and 101 years. Standard carotid artery stiffness was assessed by measuring the typical carotid *DC*. The presence of plaques was assessed by evaluating the ultra-sonographic images of the common, internal, and bifurcation sites of the carotid artery for the presence of atherosclerotic lesions. A total carotid plaque score was defined by summation of the presence of plaques at far and near walls of the left and right sides at three locations. Severity was graded as no plaques (score 0), mild plaques (score 1 to 4), moderate plaques (score 5 to 8), and severe plaques (score 9 to 12). The authors have found a relation between the severity of plaques and the measured arterial stiffness: as the presence of plaques in the common carotid artery becomes more severe, the *D* value decreases. Although there was no significant difference between the groups classified as no plaque (10.9 × 10^−3^ kPa^−1^), mild plaque (10.8 × 10^−3^ kPa^−1^), and moderate plaque (10.1 × 10^−3^ kPa^−1^), the group of patients with severe plaques presented a significantly different and much lower value of *DC* (8.7 × 10^−3^ kPa^−1^). Therefore, the authors have concluded that arterial stiffness is strongly associated with atherosclerosis at various sites in the vascular tree [44].

On the other hand, London et al. (2003) [45], by measuring the PWV, have studied the correlation between arterial stiffness and arterial intimal calcification as a consequence of atherosclerotic plaque development. The investigation was done in 202 patients that underwent B-mode ultra-sonography of the common carotid artery to determine the presence of atherosclerotic calcified plaques. The authors found significant differences between healthy arteries and calcified arteries. While non-calcified arteries had a PWV of 921 ± 176 cm/s, arteries with intimal calcifications presented a PWV value of 1056 ± 230 cm/s, which means that calcification deposits in arteries increase the arterial stiffness of the blood vessel [45]. Nevertheless, due to the heterogeneity of plaque composition, the association between arterial stiffness measured by PWV, and the presence of atherosclerosis has been inconsistent [43].

Table 3 lists results from the reviewed studies regarding dynamical arterial changes during the cardiac cycle. Although there are some discrepancies between results from different studies, it is noted that an overall conclusion is the sense that decreased arterial elasticity (or increased stiffness) is associated with atherosclerotic plaque presence [20].

Uniaxial tensile tests

Literature regarding mechanical tests of atherosclerotic plaques via tensile tests is very limited. Moreover, the variability of techniques used to perform such mechanical tests is so widespread that comparisons between atherosclerotic plaques, even from the same vasculature, are very hard [40,46]. Nevertheless, some of the studies that report the mechanical behavior of atherosclerotic blood vessels under tensile tests are presented in this paper. Most of them focused on arterial tissue.

In uniaxial tensile tests, the sample is elongated in a single direction, and the stress induced in the tested specimen (in this case, biological tissue) is quantified. The most frequently used stress measures are the Cauchy (or true) stress (σ) and the engineering (or first Piola–Kirchhoff, or nominal) stress (*P*). Both measurements are obtained from the ratio between the applied force (*F*) and a cross-sectional area. Cauchy stress assesses the changes in the current cross-sectional area (*A*), whereas the engineering stress considers the reference (original) cross-sectional area (*A_0_*) [40].

To measure the non-linear strain of plaques, the most used parameter are the stretch ratio (*λ* Equation (10)) and the nominal strain (*ε* Equation (11)) [40]. Both measurements evaluate the length changes, following the variation from the reference (original) length (*L_0_*) to the deformed length (*L*).
(10)$$\lambda =\frac{L}{L_{0}}$$
(11)ε=λ−1 

Bailey, in 1965, [47] was one of the first researchers to study the correlation between mechanical behavior and visual intensity of atherosclerotic plaques using a commercially available textile load-testing machine. The author used isolated strips of normal and atherosclerotic rabbit aorta to determine elasticity, breaking strain, and other tensile parameters. Atherosclerotic plaques were induced by feeding rabbits with a cholesterol-rich diet. Results indicated no significant decrease in tensile strength (breaking strain) of the tissue with increasing severity of atherosclerosis. This fact was explained by the fact that early stages of atherosclerosis affect primarily the intima layer and not the media and adventitia layers, which contribute the most to the tensile strength of the tissue.

Regarding Young’s modulus in the elastic stretching domain, it was noted that increased atherosclerosis severity presented decreased elasticity. Inelastic stretching, in turn, was found to decrease between the first (“unstressed”) and subsequent (“prestressed”) tests, presumably meaning that the first breaking stress produces irreversible changes not restored in the interval before the next test. For both types of inelastic stretching, no significant difference between normal and atherosclerotic aortas was found [47]. Bailey has not extracted definitive conclusions about this study.

In the study by Maher et al. (2009) [46], the authors have characterized the mechanical behavior of fresh human carotid plaques removed during endarterectomy. Fourteen carotid plaques were collected, each classified as calcified, mixed, or echolucent, through duplex ultrasound. The authors performed 17 circumferential tensile uniaxial tests. Results have shown that calcified plaques had the stiffest response, while echolucent plaques were the least stiff. They have also found differences in the behavior of samples taken from different anatomical locations (common, internal, and external carotid).

Lawlor et al. (2011) [48] analyzed atherosclerotic plaque characteristics from 18 patients tested on-site, with post-surgical revascularization through endarterectomy. Using the ultrasound technique, each sample was classified as calcified, mixed, or echolucent. The ultimate tensile strength of each sample was performed through the uniaxial tensile test in the circumferential direction. It showed considerable variation across the 14 atherosclerotic samples tested: strains at rupture varied from 0.299 to 0.588, and the Cauchy stress observed in the experiments was between 0.131 and 0.779 MPa. No significant difference in mechanical behavior between plaque types was found.

In the study reported by Loree et al. (1994) [49], the stress–strain behavior of 26 human aortic intimal plaques was studied. Plaques were removed during routine autopsies of 21 patients from their aortas (abdominal and thoracic), and they were histologically classified as cellular, hypocellular or calcified. The uniaxial tensile test was performed in the circumferential direction. At a physiologic applied circumferential tensile stress of 25 kPa, the tangential moduli of cellular, hypocellular, and calcified specimens were 927 ± 468 kPa, 2312 ± 2180 kPa, and 1466 ± 1284 kPa, respectively.

Cunnane et al. (2016) [35] have characterized the stretch at failure, strength, and stiffness of human atherosclerotic plaques using circumferential planar tensile tests. Plaques were excised from the carotid and femoral vessels of 40 patients that underwent carotid or endarterectomy to treat high-grade arterial stenosis. After being collected, plaque samples were spectroscopically characterized through FTIR, and composition characterization was made from the analysis of the peaks of interest. They have observed that, while atherosclerotic plaques obtained from the femoral vessel were predominantly fibrocalcific, the ones extracted from carotid vessels were mainly constituted by lipid core atheroma. Femoral plaques presented a stretch at the failure of 1.62 ± 0.27 and strength and stiffness of 0.22 ± 0.12 MPa and 0.44 ± 0.26 MPa, respectively. In turn, carotid plaques showed strength at the failure of 1.89 ± 0.36 and strength and stiffness of 0.49 ± 0.23 MPa and 0.89 ± 0.51 MPa, respectively. From the obtained results, authors have found a negative correlation between calcified plaques and stretch at failure and a positive correlation between lipidic plaques and this mechanical property. These findings confirm that the calcification reduces the load-bearing capacity of diseased arterial tissue, whereas high levels of lipid content increase tissue strain under physiological loading [35].

In turn, the main goal of the study performed by Kobielarz et al. (2020) [2] was to determine experimentally under uniaxial tensile loading the mechanical properties of different types of human aortic atherosclerotic plaques. A total of 37 different plaques were identified as predominantly calcified, lipidic or fibrotic based on vibrational spectra analysis and validated by histological staining. Results showed that fibrotic plaques were stiffer than the normal aorta, presenting a stiffness for the high stretch domain value (λ_H_) of 8.15 MPa and 6.56 MPa for circumferential and axial directions, respectively. Fibrotic plaques presented a stress at failure value (σ_M_) of 1.57 MPa and 1.64 MPa for circumferential and axial directions, respectively, showing a stronger behavior compared to normal aortas. In contrast, lipidic plaques were the weakest of all types of plaques (σ_M_ of 0.76 MPa and 0.51 MPa for the circumferential and axial directions, respectively); and the calcified plaques were the stiffest (λ_H_ of 13.23 MPa and 6.67 MPa for circumferential and axial directions, respectively). Authors have concluded that the averaged stretch and stress at failure decrease as the calcification content increases, and a decrease in the number of fibrotic elements leads to a decrease in load-bearing properties. On the other hand, fibrotic plaques were characterized by higher stiffness, higher stress at failure, and lower stretchability [2].

Compression tests

Compression tests are also valuable mechanical tests to characterize atherosclerotic blood vessels. Although in a physiological situation, atherosclerotic blood vessels experience circumferential stretching during blood pressure pulsation, these tissues also sense radial compression during this pulsation event. Therefore, besides evaluating the mechanical behavior of plaques through tensile tests, compression tests are physiologically relevant [50].

In unconfined compression tests, samples are mounted between two metal plates, of which the bottom plate can be raised and lowered with a pre-set speed to pre-set positions, and the top plate is stationary and attached to a load measuring device. In this mechanical test, displacement and force can be measured as loading conditions [50].

Micro-indentation technique can also be used to measure the material stiffness of tissues. An indenter is impressed into the biological material, and an indentation load-depth curve is obtained. From this curve, mechanical parameters, such as Young’s modulus, hardness, residual stress, and fracture toughness, can be determined [51]. However, due to the variation in tissue thickness, tissue stiffness cannot be directly derived from load-depth curves, so numerical simulations are necessary to extract mechanical information [50].

For instance, Maher et al. (2009) [46] have not only evaluated the mechanical behavior of atherosclerotic plaques in terms of circumferential tensile behavior, but they also have characterized those tissues in terms of radial compressive behavior. Specimens were removed from the human carotid artery at the bifurcation site and separated into common, internal, and external carotid segments. Plaque classification was determined independently by a clinician using routine Duplex ultrasound with greyscale imaging, and they were classified as calcified, mixed, or echolucent. Unconfined compression tests were performed in the radial direction at a compressive rate of 1%/s until 60% strain was reached. Samples showed a nonlinear behavior, which was under the expectation. Although authors have observed high variability in the compressive behavior for both calcified and mixed plaques, from the mean curves of each classification type, they have concluded that calcified plaques were, on average, over twice as stiff as the echolucent samples, and 1.5–2 times stiffer than the mixed plaques. Calcified plaques showed an average tangential stiffness of 140 kPa at 5% compression and 2300 kPa at 20% compression, whereas echolucent and mixed plaques showed an average stiffness of 20 kPa and 20 kPa at 5% compression and 100 kPa and 330 kPa at 20% compression, respectively. However, they could not find a relation between the anatomical location of the plaque and its mechanical behavior [46].

Lee et al. (1992) [52] studied the biomechanical properties of 43 atheroma caps obtained from the abdominal aortas of 22 human patients. Specimens were classified as non-fibrous, fibrous, or calcified based on intravascular ultrasound appearance. Mechanical characterization was performed via a uniaxial unconfined compression test. An initial compressive stress of 4.0 kPa was applied until a static equilibrium was reached. Then, the compressive stress was increased to 12 kPa, and strain and creep times were recorded for this step. By measuring the static strain caused by increasing levels of compressive stress, the static stiffness (ratio of stress to strain) was determined. The times to reach static equilibrium (creep time) for the non-fibrous, fibrous, and calcified classes were 79.6 ± 26.5 min, 50.2 ± 20.0 min, and 19.4 ± 8.1 min, respectively. The compression strain was 24 ± 11% for non-fibrous caps, 11 ± 5% for fibrous caps, and 3 ± 2% for calcified caps. The static stiffnesses of the non-fibrous, fibrous, and calcified classes were 41.2 ± 18.8 kPa, 81.7 ± 33.2 kPa, and 354.5 ± 245.4 kPa, respectively [52].

Topoleski et al. (1997) [53] have investigated the radial compressive behavior of aortic iliac plaques by quasi-static compressive test. Samples underwent a complex multiple-cycle compression protocol consisting of two 15-cycle loading phases separated by a 10–15 min unloaded rest period. Stress and stretch were calculated from the load and optical-encoder-displacement data. The Lagrangian stress was determined by dividing the instantaneous load by the original cross-section area of the specimen, and the stretch was defined as the current thickness divided by the original individual specimen thickness. Plaques were obtained during autopsies and divided into three types based on mechanical behavior. Differences in behavior were associated with histological features. All three types of plaques showed distinct mechanical behavior regarding repeatability and recoverability. The maximum compressive stretch at 350 kPa loading stress was 86 ± 9%, 46 ± 9%, and 30 ± 11% for calcified, fibrous, and atheromatous samples, respectively, and 30 ± 6% for healthy tissues. From the obtained results, authors have found that fibrous and atheromatous plaques were less stiff than calcified samples, as expected [53].

A micro-indentation device was used to study the quasi-static radial compressive properties of fibrous caps extracted from carotid artery plaques. In turn, Barret et al. (2009) [54] have studied the compressive mechanical properties of 11 carotid atherosclerotic plaques using the micro-indentation technique. The elastic properties of the materials were estimated by fitting the measured indentation response to finite element simulations. Indentations were carried out until a force of 0.2 N was reached or when an indentation depth of 0.5 mm was exceeded. Tangential stiffness values were determined and ranged from 21 to 300 kPa, with a median of 33 kPa at 5–20% compression [54].

Chai et al. (2013) [55] have also studied the compressive mechanical properties of human carotid atherosclerotic plaques using the micro-indentation technique in the axial direction. Plaques were obtained from endarterectomy. Samples were tested with a 2 mm diameter spherical indenter, and the collagen fiber structure of the samples was visualized during tests using an inverted confocal microscope. Using an inverse finite element approach and assuming isotropic neo-Hookean behavior, the corresponding Young’s moduli were found in the range from 6 to 891 kPa, with a median of 30 kPa, at 30% compression. The authors have found that collagen-rich locations were stiffer than locations that were collagen poor. In addition, they concluded that Young’s moduli of structured and unstructured collagen architectures were not significantly different. Finally, they have observed that compressive mechanical properties of atherosclerotic plaques are independent of location within the plaque [55].

Table 4 lists the results of mechanical characterization studies done in atherosclerotic tissues. All studies reveal a large variation in the mechanical properties of atherosclerotic blood vessels in stiffness results. Stress–stretch behavior of the plaque cap is nonlinear, viscoelastic, composition-dependent, and different under tension and compression [53]. Therefore, it is impossible to draw definitive conclusions about the mechanical behavior of different types of plaques, even if they originated from the same vascular territory.

## 4. Additive Manufacturing to Reproduce Atherosclerotic Blood Vessels

Constricted or occluded blood vessel segments, such as stenotic or atherosclerotic vessels, are often treated through the clinical implantation of endovascular stents [56]. Usually, such endovascular interventions are supported with image-guided procedures, as they intend to be the less invasive as possible. Minimal-invasive procedures are advantageous tools inducing fewer patient complications and faster recoveries [56]. Therefore, arterial phantoms with tunable anatomical and biomechanical properties are highly demanded experimental tools for developing and validating new instruments, imaging systems, and protocols. Moreover, phantoms generated from a patient’s medical imaging data sets can serve as personalized models for diagnostic and treatment planning purposes, enabling personalized medicine [57]. 

These engineered biomodels can also be utilized to study the biophysical and biomechanical phenomena in atherosclerosis development, validate numerical studies, and complement or confirm results from in vivo experiments [58]. Engineered vascular biomodels can overcome limitations related to in vivo experimentation, such as expensiveness, low reproducibility, and ethical issues [58]. However, these phantoms must be easily reproducible and have accurately specified geometries to achieve precise training performance and reliable experimental results. This can be effortlessly accomplished with AM.

Initially developed in the 1980s, additive manufacturing, commonly known as “3D printing,” has become widely used in different areas, including medicine and healthcare [59]. Its use allows manufacturing of 3D objects as the result of a consecutive combination of 2D layers, which are slices from a digital file representing the object. As Guarnera et al. (2021) [59] wrote, “this layer-by-layer creation from a digital file enables creating complex geometries from various materials, including thermoplastic, metal, elastomers, and biomaterials”. The main advantage of 3D printing is obtaining custom-made models at a relatively low cost, with no expensive molds or casts required [60]. This is particularly beneficial to physicians as they can use those models (derived from 3D reconstructed images) for surgical planning, education, and training [61].

According to ASTM/ISO 52900:2021, AM can be categorized into seven processes [62]. The most used processes for polymeric materials are material jetting (e.g., polyjet technology), material extrusion (e.g., fused filament fabrication—FFF), vat photopolymerization (e.g., stereolithography—SLA), and powder bed fusion (e.g., selective laser sintering—SLS) [60]. The choice of adequate technology relies on the engineering requirements of the final product, namely its resolution, material properties, and cost. For instance, when printing biomimetic parts or surgical objects, the accuracy of the geometries is a highly relevant factor [59].

SLA is the most frequently used technology where a photosensitive polymer resin or hydrogel is polymerized due to the irradiation with suitable wavelengths, often in the UV range, they crosslink, giving rise to a thermosetting polymer layer-by-layer (additively). Although this strategy allows high resolution and relatively quick printing, the need for photosensitivity monomers limits the range of materials used in SLA. Additionally, thermal processing of the photosensitive material is often required, being a barrier to low-cost printing. However, SLA is still the easiest method to print surgical models quickly and accurately [60].

Alternatively, in SLS, a powder solidifies into layers upon the irradiation with a CO2 laser. Over time, the repeated layers of solidified powder will originate the desired object. Since large amounts of materials can be used in SLS, this 3D printing technology is very attractive in specific circumstances. However, despite having high accuracy and resolution, the porosity of resulted objects is generally undesired, making it an expensive and time-consuming technique because of the required post-treatment [59].

On the other hand, FFF technology does not involve any irradiation process or binding substance; instead, it consists of a filament extruded and deposited by a nozzle onto the printer’s base. This AM technology also minimizes waste material and reduces the associated cost because instead of solidifying the material (e.g., resin or powder) in a base or tray, FFF will just utilize the needed material by depositing it. Nevertheless, the materials must be “printable”, meaning that only certain materials can be deposited; nevertheless, it is the technology with the more extensive library of thermoplastic polymers available [60].

Finally, the polyjet uses an inkjet print head that jets photopolymerizable polymer droplets, which are then cured with UV radiation. As Guarnera et al. (2021) [59] wrote, since “multiple photopolymers can be deposited prior to the UV exposure, polyjet is a rare example of a commercial printer able to manufacture multi-material parts in a single print”. This means that polyjet printers can fabricate a large variety of materials with high resolution and at a low cost. Moreover, the polyjet approach enables the combination of different types of materials, combining soft rubber-like polymers with stiffer polymers and allowing printing materials more similar to biological tissues [63].

### 3D-Printed Atherosclerotic Blood Vessels

As an auxiliary tool to the cardiovascular medical field, AM has emerged in the last decades and has already demonstrated its potential. The variety of current and developing uses for the treatment of vascular disease includes the creation of models for education and training [63,64], surgical planning [65,66], and vascular device and tissue engineering [67].

Nevertheless, the literature still needs to be expanded regarding the creation of vascular models oriented to atherosclerosis disease modeling, even though some research groups have already tried to reproduce atherosclerotic blood vessels by AM techniques. In fact, Ahn et al. (2018) [68] said that “it is hard to mimic atherosclerotic plaque by 3D printing technology because atherosclerotic plaque has complex composition”.

In order to understand the relationship between spatial characteristics and hemodynamic variations, Yang et al. (2017) [69] performed hemodynamic analysis by computational fluid analysis in patient-specific stenotic vessel models and the SLA technique to produce a 3D printed model of those stenotic vessels. The 3D printed models were obtained from a printer with 75 μm of resolution with a smooth surface, capable of forming the precise replica of patient-specific coronary arterial models with a 1:4 scale. Models were printed with liquid resin, and the excess raw materials were manually removed. The authors found that 3D printed models not only facilitated the sensory understanding of the patient-specific left coronary arterial’s spatial characteristics but also enabled a reliable visualization of the stenoses severities. Additionally, they concluded that multiple spatial characteristics could be an index of hemodynamic significance. Since 3D printed models have provided accurate replicas of the patient-specific left coronary arterial trees, they can help understand the spatial distribution of the stenosis and could be an advantage for educating and preparing medical strategies [69]. 

Friedrich et al. (2020) [56] have investigated the feasibility of 3D printing for vessel phantoms with user-defined stenoses made of elastic materials. Synthetic stenosis phantoms of different geometries were modeled with computer-aided design (CAD) software. Two exemplary stenosis geometries (asymmetric and symmetric stenosis) were 3D-printed (5 cm in length, 6 mm of inner diameter, and wall thickness of 0.4 mm) with a commercially available SLA printer using an “elastic resin” provided by the manufacturer of the printer, which has a Shore hardness of 50A and silicone like appearance and seems to be a good candidate for elastic vessel phantoms. While in asymmetric geometry, stenosis is in just one side of the vessel with a spherical shape that occludes 50% of the vessel’s cross-sectional area, symmetric geometry features a rotationally symmetric hourglass-shaped stenosis that occludes 75% of the vessel’s cross-sectional area. The authors found that the transparent appearance of the printing material allowed for visual feedback during a simulated procedure, which is an advantage for testing a clinical intervention. Additionally, as the design freedom gained by AM is large, printed phantoms can be easily adapted to the requirements of the study, allowing to make reproducible tests with precisely patient-specific geometries [56].

On the other hand, Carvalho et al. (2020) [58], developed a hemodynamic study in idealized stenotic and healthy coronary arteries by a high-speed video microscopy technique. Experimental flow studies were performed in biomodels with three different resolutions (50, 100, and 150 µm) obtained from SLS 3D printing technology. Biomodels were designed in an online platform and custom-manufactured using a rigid material. Although it was a simplified model, the selected dimensions of the 3D printed biomodel allowed for obtaining good enough results to validate numerical results. The stenosis length (6 mm) was defined as twice the inlet diameter (3 mm), and the models’ total length was defined as 50 mm. Different degrees of stenosis were considered: 0% (healthy model), 50%, 60%, 70%, and 80%, corresponding to the diameter reduction at the stenosis throat. The authors concluded that the biomodel printed with a resolution of 50 µm, due to its lowest roughness values, was able to give more accurately results and precise flow visualization. They also found good agreement between experimental flow results and blood flow numerical data [58].

After concluding the best biomodel’s parameters, Carvalho et al. (2021) [70] continued their work and did a hemodynamic study in 3D printed stenotic coronary artery models to evaluate the influence of stenosis degree in blood flow distribution. The selected dimensions of the 3D printed biomodel were the same as those used in previous work [58] (length of stenosis of 6 mm, inlet diameter of 3 mm, and model’s total length of 50 mm). Different degrees of stenosis were also considered: 0% (healthy model), 50%, 60%, 70%, and 80%, corresponding to the diameter reduction at the stenosis throat. The 3D biomodels were designed in an online platform and manufactured by using an SLA printer, with a printing resolution of 50 µm (the material was not specified). Besides validating numerical calculations with 3D printed biomodels, Carvalho and co-workers have also concluded that stenosis degrees higher than 50% create disturbed flow downstream of the contraction [70]. This work provided a pathophysiologic study about atherosclerosis’s effect on blood vessels’ hemodynamic performance without using in vivo experiments (which are expensive, time-consuming, and have ethical issues) but with accurate results that can be transposed to human physiology.

To produce a promising platform to elucidate the pathophysiology of atherosclerosis and seek effective drugs and therapies, Gao et al. (2021) [25] constructed an atherosclerotic model via a novel fabrication strategy. Atherosclerotic biomodels were developed using a 3D in-bath coaxial cell printing technique resulting in a triple-layered artery equivalent model with tunable geometries. The fabrication process involved the 3D printing of a model house, deposition of bath material, coaxial cell printing of dual-layered tubes, and pump connection. The study reconstructed a native atherosclerotic environment involving co-culture cells and local flow signaling [25]. This work aimed to study atherosclerosis’s pathophysiology from a biochemical point of view instead of the biomechanical characterization, which is the objective of this review paper.

On the other hand, Guarnera et al. (2021) [59] developed a mimicking model of the external iliac artery affected by atherosclerotic plaque, employing the polyjet multi-material technique to be mechanically tested and to validate different numerical models. Polyjet print technology was chosen for its ability to print multiple materials within a single part at the required dimensions (10 mm × 40 mm × 10 mm) and resolution. In fact, 3D printed model contained six regions with distinct mechanical responses. To accomplish that, five digital materials (materials that result from the combination of soft rubber-like polymers with stiffer polymers) were produced by polyjet printing. The digital materials are mixtures obtained through jetting of two different materials: high flexible photopolymer with the ability to undergo large deformations (Agilus30) and a rigid photopolymer (VeroClear^®^). As the percentage of VeroClear^®^ increases, the hardness of the material also increases. The geometry of the 3D printed biomodel was based on the cross-section of the human external iliac artery affected by atherosclerotic plaque obtained from a high-resolution magnetic resonance image discretized through the Carrera Unified Formulation model according to mechanical properties of atherosclerotic components. To mimic adventitia, calcification deposit, fibrous cap, fibrotic media, media, and lipid pool, the authors utilized digital materials with Shore A hardness values of 50A, 95A, 50A, 70A, 40A, and 30A, respectively. Different Shore A hardness values are obtained by varying the composition of the digital materials. Mechanical characterization of each type of digital material was performed through quasi-static tensile testing, and authors found that the stiffness values of the materials were three times greater than those estimated in biological tissues [59]. Although this study was a scientific advance in atherosclerotic plaque modeling, additional work must be done to produce a more accurate biomodel of an atherosclerotic blood vessel in biomechanical properties. Future biomodels must have discretized regions, each one corresponding to a different plaque component and with mechanical properties similar to the biological tissues that are mimicking.

More recently, Song et al. (2022) [71] created an experimental model of real-structure carotid arteries with stenosis caused by atherosclerotic plaques, applying 3D printing technology. This work aimed to obtain a 3D printed model based on real configuration data to simulate the stenotic blood flow caused by carotid plaques. Flow field characteristics of carotid artery stenosis were revealed through full velocity field measurements. Then, numerical simulation calculations were done to confirm and validate physical experimental data. The physical experimental model was converted from a 3D geometric model of carotid arteries with plaques reconstructed from computed tomography (CT) images. The 3D printed model had a layer thickness of 0.1 mm and was achieved by the SLA technique using a transparent photosensitive resin. The authors concluded that 3D printed models could support the understanding of carotid artery stenosis flow characteristics accurately, and that numerical simulation was a reliable method for studying the blood flow under stenotic conditions [71].

Finally, Wu et al. (2022) [57] developed a novel ink for micro-extrusion printing realistic arterial phantoms. The ink was composed of poly(vinyl alcohol glycidyl methacrylate) (PVAGMA) and cellulose nanocrystal (CNC). Since poly(vinyl alcohol) (PVA) materials are typically strain-softening, the authors implemented CNC as a nanofiller to improve the strain-stiffening behavior of the phantoms, resembling the mechanical properties of human soft tissues. CNC was also used to enable ultrasound imaging of phantoms due to its superior compatibility with ultrasonography. Regarding obtained phantoms, the authors found that by varying the molecular ratio between PVA and glycidyl methacrylate (GMA), the mechanical properties of phantoms could be tuned, with tensile modulus presenting a wide range of values, ranging from 21.80 ± 6.59 to 146.52 ± 22.61 kPa. The phantoms revealed excellent stability and durability. Additionally, the burst pressure of materials was comparable to the native carotid artery, meaning that these materials could be good candidates to fabricate arterial vessels [57]. Although the authors have created a biomaterial to be successfully applied in producing arterial phantoms, atherosclerosis disease was not approached in this work, and atherosclerotic tissue components were not mimicked.

Regarding the use of 3D printing technologies to produce cardiovascular grafts, Melchiorri et al. (2016) [72] fabricated a tubular-like scaffold (inner diameter of 1 mm and 150 µm of wall thickness) to be used in repair and/or reconstruction of vessels in venous system. The grafts were designed to fit the specific curvature of the patient’s anatomy and were obtained through digital light stereolithography (DLP) using a biocompatible and photocrosslinkable polymer—poly(propylene fumarate) (PPF). Besides the maintenance of the mechanical properties, 3D printed models revealed no side effects in terms of thrombosis, aneurysms, and stenosis, after being implanted in mice venous system for six months. The authors have accomplished vascular grafts with sustained patency and functionality [72]. 

With a similar purpose, Sohn et al. (2021) [73] produced an artificial vascular graft using PVA filaments and FFF technology (printer nozzle with a diameter of 0.4 mm and printing speed of 60 mm/s). The authors found that these artificial grafts cause decreased thrombogenesis and increased endothelization in transplanted rats [73]. Even though these studies constitute significant advances in the cardiovascular area, they do not take in consideration the atherosclerosis disease, which is the main aim of this review.

Table 5 summarizes the most used materials in 3D printing of blood vessels, although most of the works do not specify the chemistry of the polymeric material. Moreover, some critical reviews on 3D printing and polymers to reproduce biological tissues environments can also be consulted, such as those of references [74,75].

## 5. Conclusions and Future Perspectives

This review showed:AM has the potential to revolutionize the future of many scientific fields, namely medicine, standing as an essential part of treatment plans. The capability of bioprinting complex organs or even producing 3D-printed objects to be used in training and education or applied in research fields like pharmaceutical areas and drug discovery makes AM an emerging biomedical tool;The production of biomimicking models of diseased blood vessels is an arousing demand to help and guide the treatment of vascular diseases, such as atherosclerosis;Inexistence of standardized tests to mechanically characterize biological tissues hampers the obtention of consensual values regarding the mechanical properties of blood vessels (healthy and diseased), leading to a wide range of results;A priority request is to standardize the tests used in (bio-)mechanical characterizations of blood vessels;Inconsistent mechanical properties of blood vessels and atherosclerotic plaques impede the selection of adequate AM technique and, consequently, appropriate “printable” materials to reproduce pathophysiological properties of such biological tissues;Complexity of atherosclerotic plaque composition and the variability of samples to be tested (vascular source, stage of plaque development, variability between subjects, etc.) hinder the 3D printing of biomodels capable of reliably mimicking the atherosclerotic blood vessels;The creation of 3D-printed vascular models that precisely mirror atherosclerotic vessels will be more straightforward after accomplishing consensual (or absolute) values of the mechanical properties of blood vessels.

Accordingly, AM could help understand the pathophysiology of atherosclerosis disease and its biomechanical and biophysical consequences and discover new therapeutical strategies and/or new and more effective drugs.

## Figures and Tables

**Figure 1 polymers-15-00480-f001:**
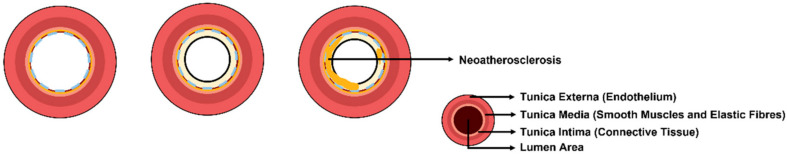
Schematic of the cross-section of a blood vessel and the formation of an atherosclerotic plaque (adapted from [6]).

**Table 1 polymers-15-00480-t001:** Mechanical properties of blood vessels wall’s components (elastin, collagen, and SMCs) and some blood vessels (adapted from [7,9]).

Human Vascular Tissue	Compliance (%/100 mmHg)	Burst Pressure (mmHg)	Young’s Modulus (MPa)	Stress at Break (MPa)	Strain at Break (%)	Ref
Elastin	NA	NA	0.3–1 [13] 0.88–1.14 [14] 0.038 ± 0.004 [15]	1–2	100–150	[13]
Collagen	NA	NA	1000	120	13	[13]
SMCs	NA	NA	0.1–2	NA	NA	[13]
Femoral Artery	5.9 ± 0.5 [16]		9–12	1–2	63–76	[17]
Internal mammary artery	11.5 ± 3.9 [18]	2000	8.0 ± 3.0 (circ) * 16.8 ± 7.1 (axial) *	4.1 ± 0.9 (circ) 4.3 ± 1.8 (axial)	134 ± 28 (circ) 59 ± 15 (axial)	[19]
Saphenous Vein	4.4 ± 0.8 [16]	1680–3900	4.2 ± 3.3 (circ) * 23.7 ± 15 (axial) *	1.8 ± 0.8 (circ) * 6.3 ± 4.0 (axial) *	242 ± 89 (circ) 83 ± 19 (axial)	[19]

* (circ) and (axial) are the tensile tests performed in the circumferential and axial directions, respectively. Stress–strain curves were obtained in room temperature using a uniaxial tensile tester. NA—not available.

**Table 2 polymers-15-00480-t002:** Parameters used to express local elasticity and stiffness of blood vessels, as a function of changes in diameter or area of those vessels during cardiac cycle (systolic and diastolic). Adapted from [20].

Stiffness Elasticity Measurement	Standard Formula		Units	Notes
Absolute distension	$\Delta D=D_{s}-D_{d}$	(1)	μm	Decreasing values indicate stiffer arteries
	$\Delta A=A_{s}-A_{d}$	(2)	${\mu m}^{2}$	
Strain	$Strain=\frac{\Delta D}{D_{d}}$ or $\frac{\Delta A}{A_{d}}$	(3)	%	Normalizes distension to the diastolic diameter (or area)
Peterson’s elastic modulus (pressure-strain elastic modulus)	$E_{p}=\frac{\Delta P}{Strain}=\frac{\Delta PD_{d}}{\Delta D}$	(4)	kPa	Divides pulse pressure ΔP by strain. Increasing values indicate stiffer arteries
Distensibility	$D=\frac{1}{E_{p}}=\frac{\Delta D}{\Delta PD_{d}}\;$	(5)	${\mathrm{kPa}}^{-1}$	Inverse of $E_{p}$. Decreasing values indicate stiffer arteries
Distensibility coefficient (area)	$DC=\frac{\Delta A}{\Delta PA_{d}}$	(6)	${\mathrm{kPa}}^{-1}$	Decreasing values indicate stiffer arteries
Young’s elastic modulus	$YEM=\frac{D_{d}}{h_{o}DC}=\frac{\Delta P{D_{d}}^{2}}{2h_{o}\Delta D}$	(7)	kPa	Accounts for vessel wall thickness ($h_{o}$). Increasing values indicate stiffer arteries
β-stiffness index	$\beta =\frac{\ln\left(\frac{P_{s}}{P_{d}}\right)}{strain}=\ln\left(\frac{P_{s}}{P_{d}}\right)\frac{D_{d}}{\Delta D}$	(8)	Adimensional	Increasing values indicate stiffer arteries
Pulse wave velocity	$PWV=\sqrt{\frac{V\times \Delta P}{\rho \times \Delta V}}$	(9)	cm/s	Increasing values indicate stiffer arteries

Legend: $D_{d}$ and $D_{s}$ are diastolic and systolic diameters, respectively; $A_{d}$ and $A_{s}$ are the cross-sectional areas of lumen in diastole and systole, respectively; $h_{o}$ is the vessel wall thickness; ρ is blood density; V is the initial arterial volume; and ΔV is the volume change.

**Table 3 polymers-15-00480-t003:** Results of studies investigating arterial stiffness in atherosclerotic vessels, from dynamic arterial changes measurements.

Reference	Type of Sample	Stiffness or Elasticity Measurement	Classification	Result
[44]	Human common carotid artery	Distensibility (×10^−3^ kPa^−1^)	No plaque	10.9 *
			Mild plaque	10.8 *
			Moderate plaque	10.1 *
			Severe plaque	8.7 *
	Human abdominal aorta artery	Distensibility (×10^−3^ kPa^−1^)	No plaque	10.6 *
			Mild plaque	10.5 *
			Moderate plaque	10.4 *
			Severe plaque	9.8 *
[45]	Human common carotid artery	Pulse wave velocity (cm/s)	Non-calcified arteries	921 ± 176
			Arteries with intimal calcifications	1056 ± 230
[42]	Human common carotid artery	Arterial strain (%)	Normal	9.4 ± 3.5
			Mild stenosis	7.7 ± 3.2
			Severe stenosis	5.8 ± 2.8
		Static pressure–strain elastic modulus (kPa)	Normal	1.85–1.95 *
			Mild stenosis	3.85–3.95 *
			Severe stenosis	6.5–7.0
[41]	Human common carotid artery	Static pressure–strain elastic modulus (kPa)	Normal	79.847 ± 7.797
			Mild stenosis	122.342 ± 15.569
			Severe stenosis	160.341 ± 45.780
		β-stiffness index	Normal	6.12 ± 0.92
			Mild stenosis	6.94 ± 0.92
			Severe stenosis	8.24 ± 0.88
		Young’s Modulus (kPa)	Normal	384.264 ± 38.986
			Mild stenosis	502.962 ± 7.784
			Severe stenosis	577.228 ± 24.156
		Distensibility (×10^−3^ kPa^−1^)	Normal	2.94 ± 0.74
			Mild stenosis	2.50 ± 1.04
			Severe stenosis	1.60 ± 0.69
		Compliance (mm^2^/Pa)	Normal	14.10 ± 1.19
			Mild stenosis	11.06 ± 0.83
			Severe stenosis	6.55 ± 0.47

Legend: * values merely graphically reported in the respective paper, and herein presented from a approximate visual estimation.

**Table 4 polymers-15-00480-t004:** Results of the studies reporting mechanical characterization of atherosclerotic tissues.

Reference	Mechanical Test	Type of Sample	Mechanical Parameter	Tissue Classification	Result
[47]	Tensile test	Rabbit aorta artery	Tensile strength (lbs.)	Normal aorta (0–5% lesions)	1.12 ± 0.38
				Moderate lesions (5–50%)	1.02 ± 0.38
				Heavy lesions (50–100%)	1.01 ± 031
			Elastic stretching (Young’s modulus) (lb^−1^)	Normal aorta (0–5% lesions)	6.60 ± 1.08
				Moderate lesions (5–50%)	6.04 ± 0.71
				Heavy lesions (50–100%)	4.40 ± 0.85
			Unstressed inelastic stretching (lb^−1^)	Normal aorta (0–5% lesions)	0.47 ± 0.16
				Moderate lesions (5–50%)	0.50 ± 0.19
				Heavy lesions (50–100%)	0.47 ± 0.19
			Prestressed Inelastic Stretching (lb^−1^)	Normal aorta (0–5% lesions)	0.23 ± 0.05
				Moderate lesions (5–50%)	0.27 ± 0.12
				Heavy lesions (50–100%)	0.29 ± 0.08
[37]	Dynamic viscoelastometry	Rabbit aortas with different levels of atherosclerotic lesions	Tangential dynamic Young’s Modulus (kPa)	No aortic lesions	290
				Mild aortic lesions	330
				Severe aortic lesions	400
[52]	Uniaxial unconfined compression test	Atheroma caps from human abdominal aorta	Creep time (time to reach static equilibrium) (min)	Non-fibrous specimen	79.6 ± 26.5
				Fibrous specimen	50.2 ± 20.0
				Calcified specimens	19.4 ± 8.1
			Compression strain	Non-fibrous specimen	24 ± 11%
				Fibrous specimen	11 ± 5%
				Calcified specimens	3 ± 2%
			Static stiffness (kPa)	Non-fibrous specimen	41.2 ± 18.8
				Fibrous specimen	81.7 ± 33.2
				Calcified specimens	354.5 ± 245.4
[49]	Uniaxial tensile test in circumferential direction	Human aortic intimal plaques	Tangential modulus at a physiologic circumferential tensile stress of 25 kPa (kPa)	Cellular specimen	927 ± 468
				Hypocellular specimen	2312 ± 2180
				Calcified specimen	1466 ± 1284
[53]	Quasi-static radial compressive test	Human aortic iliac plaques	Maximum compressive stretch at 350 kPa loading stress	Healthy tissues	30 ± 6%
				Atheromatous samples	30 ± 11%
				Fibrous samples	46 ± 9%
				Calcified samples	86 ± 9%
[5]	Tensile test	Human atherosclerotic plaques from iliac arteries	Ultimate tensile stress (kPa)	Adventitia	1031.0 ± 306.8 (c)
					951.8 ± 209.0 (a)
				Non diseased media	202.0 ± 69.8 (c)
					188.8 ± 110.9 (a)
				Non diseased intima	488.6 ± 185.6 (c)
					943.7 ± 272.3 (a)
				Fibrous cap	254.8 ± 79.8 (c)
					468.6 ± 100.1 (a)
				Fibrotic intima at the medial border	776.8 ± 336.2 (c)
					277.5 ± 98.4 (a)
				Diseased fibrotic media	1073.6 ± 289.3 (c)
					187.4 ± 8.3 (a)
			Ultimate stretch	Adventitia	1.44 ± 0.18 (c)
					1.35 ± 0.17 (a)
				Non diseased media	1.27 ± 0.08 (c)
					1.54 ± 0.33 (a)
				Non diseased intima	1.33 ± 0.2 (c)
					1.26 ± 0.15 (a)
				Fibrous cap	1.18 ± 0.10 (c)
					1.14 ± 0.07 (a)
				Fibrotic intima at the medial border	1.11 ± 0.06 (c)
					1.09 ± 0.04 (a)
				Diseased fibrotic media	1.12 ± 0.06 (c)
					1.49 ± 0.44 (a)
			Hysteresis (%)	Adventitia	17.3 ± 9.1 (c)
					13.6 ± 6.3 (a)
				Non diseased media	13.8 ± 9.8 (c)
					8.8 ± 3.1 (a)
				Non diseased intima	12.8 ± 7.1 (c)
					7.9 ± 5.4 (a)
				Fibrous cap	16.7 ± 3.9 (c)
					13.1 ± 5.2 (a)
				Fibrotic intima at the medial border	16.7 ± 3.9 (c)
					13.1 ± 5.2 (a)
				Diseased fibrotic media	5.8 ± 5.4 (c)
					4.7 ± 3.0 (a)
[46]	Unconfined compression test in radial direction	Atherosclerotic plaques from human cartid artery	Tangential stiffness (kPa) at 5% compression	Calcified plaques	140
				Mixed plaques	20
				Echolucent plaques	20
			Tangential stiffness (kPa) at 20% compression	Calcified plaques	2300
				Mixed plaques	330
				Echolucent plaques	100
[54]	Radial indentation	Human atherothrombotic tissue	Shear modulus (median in kPa) [values in brackets indicate a range]	Hypocellular atherothrombotic tissue	11 (7–100)
			Young’s Modulus (median in kPa) [values in brackets indicate a range]	Hypocellular atherothrombotic tissue	33 (21–300)
[48]	Uniaxial tensile test in circumferential direction	Human atheroscerotic plaques	Strain at rupture	Echolucent specimen	0.389–0.588
				Mixed sample	0.389–0.586
				Calcified specimen	0.299–0.474
			Cauchy stress (MPa)	Echolucent specimen	0.131–0.421
				Mixed sample	0.219–0.779
				Calcified specimen	0.366–0.606
[55]	Axial indentation	Atherosclerotic plaques from human carotid artery	Young’s Mmdulus (median in kPa)	Middle of fibrous cap	36 (SC)
					28 (UC)
				Shoulder of cap	26 (SC)
					27 (UC)
				Intima	35 (SC)
					58 (UC)
				Lipid rich necrotic core	16 (CP)
[35]	Circumferential planar tensile test	Human atherosclerotic plaques from femoral artery	Stretch at failure	Predominantly fibrocalcific specimens	1.62 ± 0.27
			Strength (MPa)		0.22 ± 0.12
			Stiffness (MPa)		0.44 ± 0.26
		Human atherosclerotic plaques from carotid artery	Stretch at failure	Predominantly constituted by lipid core atheroma specimens	1.89 ± 0.36
			Strength (MPa)		0.49 ± 0.23
			Stiffness (MPa)		0.89 ± 0.51
[2]	Uniaxial tensile Test	Human aortic atherosclerotic plaques	Stiffness for the high stretch domain (MPa)	Fibrotic plaques	8.15 (c)
					6.56 (a)
				Calcified plaques	13.23 (c)
					6.67 (a)
				Lipid plaques	4.10 (c) *
					3.50 (a) *
				Normal aorta	6.75 (c) *
					4.75 (a) *
			Stress at failure (MPa)	Fibrotic plaques	1.57 (c)
					1.64 (a)
				Calcified plaques	1.25 (c) *
					0.90 (a) *
				Lipid plaques	0.76 (c)
					0.51 (a)
				Normal aorta	1.45 (c) *
					1.35 (a) *
			Stretch at failure	Fibrotic plaques	1.32 (c) *
					1.32 (a) *
				Calcified plaques	1.13 (c) *
					1.19 (a) *
				Lipid plaques	1.32 (c) *
					1.25 (a) *
				Normal aorta	1.57 (c) *
					1.39 (a) *

Legend: (c)—obtained in circumferential direction, (a)—obtained in axial direction, (SC)—structured collagen area, (UC)—unstructured collagen area, (CP)—collagen poor area. * Values merely graphically reported in the respective paper, and herein presented from an approximate visual estimation.

**Table 5 polymers-15-00480-t005:** Materials used in additive manufacturing to reproduce vasculatures.

Material	AM Technology	Processing Methods	Observations	Is Atherosclerosis Approached?	Ref
Liquid resin	SLA	Printer resolution of 75 μm. Printed models with smooth surface. Raw material manually removed.	Better understanding about the relationship between spatial characteristics of stenotic arteries and their hemodynamic performance.	Yes	[69]
Elastic resin with silicone like appearance (provided by manufacturer)	SLA	3D printed phantoms with 0.4 mm of wall thickness.	Transparency of phantoms allows visual feedback during surgical protocol training.	Yes	[56]
Rigid material	SLS	Biomodels printed with three different resolutions and five different stenotic degrees.	Biomodels printed with better resolutions (50 µm) allow reliable results.	Yes	[58]
NA	SLA	Printing resolution of 50 µm. Biomodels printed with five different stenotic degrees.	Stenotic degrees higher than 50% disturbs the blood flow downstream the stenosis. Improved knowledge about stenosis physiology with no in vivo experimentation.	Yes	[70]
Agilus30 (flexible photopolymer) and VeroClear^®^ (rigid photopolymer)	Polyjet	3D printed models with six different regions, representing different mechanical responses.	Differences between the mechanical properties of models and mechanical properties of corresponding biological tissues are very large.	Yes	[59]
Transparent photosensitive resin	SLA	Biomodels printed with a layer thickness of 0.1 mm.	3D printed models allow better understanding about the blood flow under stenotic conditions.	Yes	[71]
PVAGMA and CAN as nanofiller	μ-extrusion	Different inks were produced by varying the molecular ratio between PVA and GMA.	Phantoms have tunable mechanical properties. Obtained materials can serve as good candidates for blood vessels transplantation.	No	[57]
Poly(propylene fumarate)	DLP	Cardiovascular grafts fitting patient’s anatomy, with inner diameter of 1 mm and 150 µm of wall thickness.	3D printed models have maintained their mechanical properties during 6 months of implantation on mice venous system. No thrombosis, aneurysm or stenosis were seen.	No	[72]
PVA	FFF	Nozzle diameter of 0.4 mm. Printing temperature of 200 °C, and printing speed of 60 mm/s.	Artificial vascular grafts have decreased thrombogenesis and increased endothelization, when compared to standard grafts.	No	[73]

NA—not available.

## Data Availability

The data presented in this study are available on request from the corresponding author.

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
