# Peer review of "Understanding Atherosclerosis Pathophysiology: Can Additive Manufacturing Be Helpful?"

_polymers, 2023, doi:10.3390/polym15030480_

Round 1

Reviewer 1 Report

The review deals with an interesting subject, which is the atherosclerosis as one of the leading causes of death worldwide. In particular, the authors presented the atherosclerosis disease and the consequences of this pathology, they discussed the mechanical characterization of atherosclerotic vessels/plaques, and they introduced additive manufacturing as a potential strategy to increase the understanding of athero-sclerosis treatment and pathophysiology.

The paper is of clear interest or the readers of the journal.

Nevertheless, the paper is not suitable for publication in the present form and some minor changes should be properly addressed by the authors to improve the clarity and the understanding.

- I would suggest the authors to shorten the first part of the review (Paragraph 1-3) to give more imprtance to paragraph 4, which is the focus and central core of the article, as also underlined by the title. In particular, I would even shorten paragraph “Additive Manufacturing to Reproduce Atherosclerotic Blood Vessels”, which is widely known. A suggestion could be make a table summarizing the main models further described, in which also the used 3D printing technology could be reported.

- In this paragraph, also considering the topic of the journal, I would suggest to give more emphasis to the employed biomaterials. Two critical reviews on a related topic, 3D printing and natural polymers, which have been employed to reproduce the vasculatures, could help

1) Hann, S.Y., Cui, H., Esworthy, T., Miao, S., Zhou, X., Lee, S.J., Fisher, J.P. and Zhang, L.G., 2019. Recent advances in 3D printing: vascular network for tissue and organ    regeneration. Translational Research, 211, pp.46-63.

2) Petta, D. et al., 2022. Musculoskeletal tissues-on-a-chip: Role of natural polymers in reproducing tissue-specific microenvironments. Biofabrication. 14, 042001.

Minor revisions

- The authors should double-check all the acronyms and describe them only the first time they appear.

- In Table 3 and 4, in the “Result” column, the measure unit is missing.

- I would suggest removing Equations 10 and 11, as they are widely known form the general readers of the journal.

- Table 4 should be moved when it is cited in the text.

I would suggest the authors to report some more figures from the cited papers. Specifically, in the section focused on 3D printing, a summarized table and a figure is missing. It would be nicer to improve the clarity and the understanding.

Author Response

The authors acknowledge the reviewer for the exceptional work reading and evaluating the manuscript. The answer to every issue is given below and, when it applies, the changes in the manuscript are highlighted in yellow.

Reviewer 1

The review deals with an interesting subject, which is the atherosclerosis as one of the leading causes of death worldwide. In particular, the authors presented the atherosclerosis disease and the consequences of this pathology, they discussed the mechanical characterization of atherosclerotic vessels/plaques, and they introduced additive manufacturing as a potential strategy to increase the understanding of atherosclerosis treatment and pathophysiology.

The paper is of clear interest or the readers of the journal.

Answer. The authors acknowledge the reviewer for the kind comments that are truly inspiring.

Nevertheless, the paper is not suitable for publication in the present form and some minor changes should be properly addressed by the authors to improve the clarity and the understanding.

- I would suggest the authors to shorten the first part of the review (Paragraph 1-3) to give more importance to paragraph 4, which is the focus and central core of the article, as also underlined by the title. In particular, I would even shorten paragraph “Additive Manufacturing to Reproduce Atherosclerotic Blood Vessels”, which is widely known. A suggestion could be make a table summarizing the main models further described, in which also the used 3D printing technology could be reported.

- In this paragraph, also considering the topic of the journal, I would suggest to give more emphasis to the employed biomaterials. Two critical reviews on a related topic, 3D printing and natural polymers, which have been employed to reproduce the vasculatures, could help

1) Hann, S.Y., Cui, H., Esworthy, T., Miao, S., Zhou, X., Lee, S.J., Fisher, J.P. and Zhang, L.G., 2019. Recent advances in 3D printing: vascular network for tissue and organ    regeneration. Translational Research211, pp.46-63.

2) Petta, D. et al., 2022. Musculoskeletal tissues-on-a-chip: Role of natural polymers in reproducing tissue-specific microenvironments. Biofabrication. 14, 042001.

Answer: This suggestion is a bit confusing because it is the authors believe that when the reviewer states Paragraph it means chapter. If this assumption is correct, and although the reviewer claim that the subjects treated in chapter 1 to 3 and even some subchapters in chapter 4 are well known, the authors must consider the different background (medical doctor, students, professors, industries) and different stages of knowledge (BSc students, MSc students and PhD students) that may be interested in this subject. Therefore, a brief knowledge of all the different aspects involving the theme must be addressed (Chapters 1 to 3). Considering that the reviewer also wanted to shorten a subchapter in 4, the authors must consider a balance between this request and the request of the other reviewer that ask exactly the opposite. Regarding the suggestion given of introducing the two references it was accepted and a table with the used biomaterials was also added.

Minor revisions

- The authors should double-check all the acronyms and describe them only the first time they appear.

Answer. The authors have made the double-check and modified the text accordingly to the reviewer suggestions.

- In Table 3 and 4, in the “Result” column, the measure unit is missing.

Answer: Because the measure unit is not the same through each table, the units were placed in column “Stiffness or Elasticity Measurement” in Table 3 and in the column “Mechanical Parameter” in Table 4.

- I would suggest removing Equations 10 and 11, as they are widely known form the general readers of the journal.

Answer: The equations were removed according with the reviewer suggestion. The text and the subsequent equations were changed accordingly.

- Table 4 should be moved when it is cited in the text.

Answer. Table 4 was moved nearest to the location where it is cited in the text.

-  I would suggest the authors to report some more figures from the cited papers. Specifically, in the section focused on 3D printing, a summarized table and a figure is missing. It would be nicer to improve the clarity and the understanding.

Answer. The authors completely agree with the reviewer that the insertion of some figures would be more clarifying. The problem is that the few figures that could be used in the manuscript must be paid, with values between 150 and 300 euros. Considering that in order to cope with the open access policies the publications are already expensive. If we must add the value paid by the figures it would become excessive. Therefore, the authors choice was to not use paid figures.

Reviewer 2 Report

The abstract is very well written, it provides all the necessary information about the present state of the art and points to a research gap that needs to be filled. Also, the introduction part is reasonably combined. The authors described the main background of atherosclerosis (its development and types of interventions) and then highlighted the main aim of their work. 
The next part is related to the description of the mechanical properties of all blood vessel components which is covered by the proper amount of literature sources. The main issues of this manuscript start with the part related to additive manufacturing which was treated in quite chaotic way. I listed all issues below:

1. 3D Printing is a colloquial version of additive manufacturing (AM), I suggest using this more academic version (AM) - especially in the title. 

2.  The authors mentioned ASTM/ISO 52900:2021 standard, but they completely missed metal technologies such as powder-based fusion (Laser and electron beam) and direct energy deposition technologies. The authors should mention them, especially since there are some works available related to i.e. stents production. I understand that this is a "Polymers" journal, but it should be mentioned. 

3. The part with AM technologies for plastics is also very generally written. Those technologies should be described in a more detailed way, especially from the biomedical point of view. There should be highlighted pros and cons of each technology in the production of biological tissues. 

4. Chapter 4.1. should be divided into subchapters related to each AM technology (i.e. Photopolymerization, Material Extrusion, etc.)

5. The should be some conclusion (listed point by point) of this work, where the authors should summarize the present state of the art and highlight the main issues in AM technologies which are properly developed now in this topic, and also point issues which need further improvements. 

After correction of all mentioned major issues the manuscript could be accepted.  

Author Response

The authors acknowledge the reviewer for the exceptional work reading and evaluating the manuscript. The answer to every issue is given below and, when it applies, the changes in the manuscript are highlighted in yellow.

Reviewer 2

The abstract is very well written, it provides all the necessary information about the present state of the art and points to a research gap that needs to be filled. Also, the introduction part is reasonably combined. The authors described the main background of atherosclerosis (its development and types of interventions) and then highlighted the main aim of their work. 
The next part is related to the description of the mechanical properties of all blood vessel components which is covered by the proper amount of literature sources. The main issues of this manuscript start with the part related to additive manufacturing which was treated in quite chaotic way. I listed all issues below:

  1. 3D Printing is a colloquial version of additive manufacturing (AM), I suggest using this more academic version (AM) - especially in the title. 

Answer. The reviewer is correct. As explained in the manuscript in abstract (lines 15-16) and within the text (in line 708), 3D printing is the common designation given to any additive manufacturing process. The change in the title was made according to the reviewer suggestion.

  1. The authors mentioned ASTM/ISO 52900:2021 standard, but they completely missed metal technologies such as powder-based fusion (Laser and electron beam) and direct energy deposition technologies. The authors should mention them, especially since there are some works available related to i.e. stents production. I understand that this is a "Polymers" journal, but it should be mentioned. 

Answer. The reviewer must consider that, within the scope of this review, only polymeric materials are considered (as stated in line 712) to mimic blood vessel and atherosclerotic plaques. Stents, devices used to overcome the medical consequences of the described biological problem, are not the focus of the manuscript. For this reason, only process or technologies related with polymeric materials are addressed (although some of them are also used for the manufacture of metallic components). For processes and technologies more specifically related with metallic materials the readers must read the document that unequivocally addresses everything in additive manufacturing: the ISO/ASTM 52900:2021 standard.

  1. The part with AM technologies for plastics is also very generally written. Those technologies should be described in a more detailed way, especially from the biomedical point of view. There should be highlighted pros and cons of each technology in the production of biological tissues. 

Answer: Once again the authors acknowledge the suggestion given by the reviewer. But, also, once again the reviewer must take in consideration that the aim of the presented manuscript is specifically the intersection between additive manufacturing and atherosclerosis pathophysiology. Therefore, describing in detail the additive manufacturing processes for polymeric materials as well as pros and cons of such processes in the manufacturing of biological tissues has been the focus of innumerous publications over the last few years. Some very few examples are given below. Consequently, it is the authors understanding that the required information is not within the aim of this review because it is not a novelty and does not contribute to the advance of the state of the art.

A Review of Three-dimensional Printing for Biomedical and Tissue Engineering Applications (2021). doi: 10.2174/1874070701812010241

Polymeric biomaterials for 3D printing in medicine: An overview (2021) doi:10.1016/j.stlm.2021.100011

Advantages of Additive Manufacturing for Biomedical Applications of Polyhydroxyalkanoates (2021). doi: 10.3390/bioengineering8020029

Additive manufacturing technology of polymeric materials for customized products: recent developments and future prospective (2021). doi:10.1039/D1RA04060J.

A comprehensive review on additive manufacturing of medical devices (2022). doi:10.1007/s40964-021-00188-0.

  1. Chapter 4.1. should be divided into subchapters related to each AM technology (i.e. Photopolymerization, Material Extrusion, etc.)

Answer: Considering the answer given to the previous point, where no detail description of the processes is going to be given, it does not seem necessary to make such sub chapters.

  1. The should be some conclusion (listed point by point) of this work, where the authors should summarize the present state of the art and highlight the main issues in AM technologies which are properly developed now in this topic, and also point issues which need further improvements. 

Answer: The authors acknowledge the suggestion. The conclusions were modified according to the reviewer comment.

Round 2

Reviewer 2 Report

The paper could be published